# Nitrile glove composition and performance— Substandard properties and inaccurate packaging information

**Ashley Herkins**[1¤a], **Sriloy Dey**[2☺], **Dan Conroy**[3☺], **Katrina Cornish**[1,4,5¤b]*

1 Department of Food, Agricultural, and Biological Engineering, The Ohio State University, Columbus, Ohio, United States of America, 2 Department of Food Science and Technology, The Ohio State University, Columbus, Ohio, United States of America, 3 Department of Chemistry and Biochemistry, The Ohio State University, Columbus, Ohio, United States of America, 4 Department of Horticulture and Crop Science, The Ohio State University, Columbus, Ohio, United States of America, 5 EnergyEne Inc., Wooster, Ohio, United States of America

☺ These authors contributed equally to this work.
¤a Current address: Department of Engineering, Boston College, Chestnut Hill, Massachusetts, United States of America
¤b Current address: United States Department of Agriculture, U.S. Arid Land Agricultural Research Center, Maricopa, Arizona, United States of America
* cornish.19@osu.edu

**Data Availability Statement:** The data underlying the results presented in the study are available from Will Ray, the designated contact for data storage for the Department of Food, Agricultural,

## Abstract

The durability and mechanical properties of synthetic medical gloves, such as those made from nitrile, vary drastically depending on the manufacturer. This study reports the chemical composition of several brands of nitrile gloves via FTIR and solid-state NMR analysis and relates composition to glove durability (found via GAD), mechanical performance (found via Instron), and whether the gloves meet or fail ASTM International standards. Out of the four nitrile examination glove brands tested, American Nitrile Slate brand had superior durability results and was found to be made of acrylonitrile butadiene rubber, as expected. The U.S. Medical glove brand, which was also found to be pure nitrile, had the superior tensile results, consistently reaching over 800% elongation before breaking. Although Restore Touch brand exam gloves were made of nitrile, they exhibited substandard tensile strength and durability due to the thinness of the glove, which barely met the ASTM minimum thickness value. The Vglove brand glove had the overall worst mechanical properties, did not meet ASTM requirements, and had an NMR spectrum consistent with that of a polyvinyl chloride glove, rather than nitrile. Gloves that fail to meet the minimum performance requirements should not be used for medical purposes to protect the health and safety of consumers.

## Introduction

Protective gloves are a first line of defense for healthcare workers and their patients, protecting against the transmission of pathogens and toxins. Acrylonitrile butadiene rubber (NBR) is a petroleum-based synthetic polymer that is widely used to manufacture examination gloves.

and Biological Engineering at Ohio State. His contact is ray.29@osu.edu. The data are also available from the corresponding author upon reasonable request.

**Funding:** This work was supported by the USDA National Institute of Food and Agriculture, OHO01524, accession 7003189. The funders had no role in study design, data collection and analysis, decision to publish, or preparation of the manuscript.

**Competing interests:** I have read the journal's policy and the authors of this manuscript have the following competing interests: Katrina Cornish is the CEO of EnergyEne, Inc. the company that lent the GAD for use in this study. The other authors declare no conflicts of interest.

Although not as durable or as comfortable as natural latex gloves [1–5] nitrile and other synthetic gloves do not induce Type I latex allergic reactions, as may occur when using improperly leached natural latex protects [6]. Residual chemicals may be present at different levels in the various glove lots which could cause contact reactions. However, it is not the purview of this paper to ensure compliance with all the FDA requirements for nitrile gloves. Because nitrile is derived from petroleum, it is also a non-biodegradable material. The COVID-19 pandemic caused an unprecedented surge in demand for nitrile gloves. Consequently, low inspection rates of these gloves resulted in the U.S. market being flooded with poorly made products in recent years [7].

Among synthetic elastomers, nitrile has gained wide acceptance as a glove that has much better performance (strength, softness and elasticity) than PVC or PE gloves, but is much cheaper than the better performing polyisoprene and chloroprene gloves. However, although nitrile gloves have improved in mechanical properties since their introduction, they are not generally viewed as a preferred glove material overall because they lack the properties of natural latex gloves and high-cost synthetics and have minimal tear resistance.

Previous durability studies have demonstrated that nitrile gloves have vastly different in-use times to failure depending on the manufacturer, and many brands failed to meet the minimum requirements specified by ASTM International (ASTM D-6319) and the U.S. Food and Drug Administration (FDA) [1, 8, 9]. This has raised serious concerns as to whether these gloves are simply poorly made, or if the nitrile has been partially or completely substituted by a cheaper, less durable alternative, such as polyvinyl chloride (PVC), or by diluent fillers like calcium carbonate. Even a tear the size of a pinhole can allow pathogenic viruses and bacteria, some of which are deadly, to transfer through the medical glove.

The purpose of this study is to evaluate a range of gloves imported into the United States as nitrile examination, emergency response technician (EMT) and industrial-grade gloves with respect to their polymer composition, durability, and mechanical properties.

## Methods

### Glove samples

This study is not intended to survey all brands and manufacturers of nitrile gloves but is an in-depth evaluation of a selection of readily available gloves in use at our institution. All gloves tested were brand new in-package and unused. The gloves that were tested had no visible holes, tears, or physical defects prior to testing. Because glove properties can vary depending on temperature, all gloves were stored and tested at room temperature (22˚C). Four brands of nitrile examination gloves (Restore Touch, U.S. Medical Glove, Vglove, and American Nitrile Slate), two brands of nitrile EMT gloves (Curaplex and Ansell Life Star), and one brand of industrial-grade nitrile gloves (N-Dex Plus) were evaluated. One brand of polyvinyl chloride (PVC) gloves (Safeko) was used as a positive control for FTIR tests. One brand of glove made from natural Hevea latex (Aloe Touch) was included in the FTIR testing as an additional reference. A solidified sample of pure nitrile latex was used as a negative control for FTIR tests. All gloves tested were a size large, with the exceptions of Safeko and Aloe Touch brands, which were both size medium. Manufacturer information: Restore Touch (Medline), $0.09/glove, Northfield, IL, USA, U.S. Medical Glove, $ 0.09/glove, Montgomery, IL, USA, American Nitrile Slate, $0.15/glove, Grove City, Ohio, USA, Curaplex, $0.25/glove, Dublin, OH, USA, Ansell Life Star, $0.29/glove, Iselin, NJ, USA, 8005PF N-Dex Plus (Showa Gloves), $0.17/glove, Menlo, GA, USA, Safeko, $0.05/glove, Brooklyn, New York, USA, Aloe Touch (Medline), $0.10/glove, Northfield, IL, USA. Vglove brand does not provide manufacturing information for their gloves, either on their packaging or online.

## Durability tests

Durability testing was performed using the Glove Durability Assessment device (GAD), which was developed to allow the user to objectively compare the durability of medical gloves without the need for manual inspection [1, 10]. Previous reports refer to this device as the New Glove Durability Assessment Device (N-GAD). Five trials were performed for each glove brand and type, with the exception of Ansell Life Star brand EMT gloves, for which four trials were performed due to limited quantities available. The 120-grit sandpaper used to create a rough glove contact surface was replaced between each trial. Although 120-grit sandpaper was selected, any grit of sandpaper could be used with the GAD, as long as the selection remains consistent for the entire study. This is because the GAD provides data on relative durability via the order of failure of glove samples, and therefore, the relative values should remain consistent regardless of sandpaper selection. It should be noted that the GAD simply presses the glove against the sandpaper–it does not drag the glove across the rough surface, which would cause abrasion of a type not encountered in normal glove use. The default settings for roller force (15 N) and speed (3.5 mm/s) were utilized for these durability tests. Following durability testing, the middle finger of each glove was removed, the thickness of the fingertip (mm) was measured three times using electronic calipers, and the median value was recorded. The average of these values was then reported for each glove variety. The thick N-Dex industrial glove was too stiff to fit onto the mandrel, and so durability could not be assessed.

## Tensile tests

Tensile data were collected according to a modified version of ASTM D412 [11]. Five dumbbells were cut out of the glove samples using Die C (CCSI, Akron, OH, USA). Tensile properties were determined using a tensiometer (model 5542, Instron, Norwood, MA, USA). The thick N-Dex glove (0.39 mm average) usually failed to break during the tensiometery tests, and so was not included in the dataset. The stress and strain data collected also provides information on the modulus of elasticity, or stiffness of the glove. The modulus of each glove type is not explicitly stated, however, for brevity.

## Fourier Transform Infrared Tests (FTIR)

A square-shaped sample with a side length of approximately 12.7 mm was cut from each glove and placed on the crystal. For the solidified pure nitrile, a thin layer was placed so that it completely covered the crystal. The pressure clamp was then used to press the sample. Spectral data were collected using an FTIR spectrometer (model 4500a, Agilent Technologies Inc., Danbury, CT, USA) equipped with a diamond-ATR accessory, ZnSe beamsplitter and DTGS detector. MIR spectra were collected over the range of 4000–700 cm$^{-1}$ with a resolution of 4 cm$^{-1}$, and 64 spectra were co-added to improve the signal to noise ratio. The infrared spectra of background and samples were recorded on a personal computer using Agilent MicroLab PC software (Agilent Technologies Inc., Danbury, CT, USA). The device was thoroughly cleansed with ethanol and wiped clean between samples to prevent cross contamination.

## Solid state Nuclear Magnetic Resonance Tests (NMR)

Glove samples were cut up and packed into 3.2 mm zirconium rotors and spun at 10 kHz MAS at 300 K. An aqueous dispersion of butadiene & acrylonitrile was packed wet into the solid-state rotor via tabletop centrifugation. Solid-state quantitative multi-cross-polarization (CP) experiments were acquired with a Bruker Avance IIIHD 600MHz (14.1T) NMR spectrometer equipped with a 3.2 mm triple-resonance (HXY) DNP probe tuned in $^1$H-$^{13}$C double

mode [12]. Quantitative multi-CP experiments with 11 ms total CP duration and 10,240 scans were calibrated on an external standard (*N*-acetyl-valine) for a total experimental time of 19.5 hrs. Recycle delays were set to 3.0 s. Spectra widths were 394 ppm with an acquisition time of 34.4 ms. Spectra were processed in Bruker TopSpin with no additional linebroadening and with a polynomial baseline correction. The acquisition settings and parameters for all spectra were identical. Note that these were solid samples. They were not dissolved in any solvent and therefore do not have a concentration to reference. No solvent suppression was used.

## Statistical analysis

Statistical analyses were preformed using the software JMP 16 and included a one-way analysis of variance test and a Tukey-Kramer HSD test.

## Results

### Durability

Curaplex and Ansell Life Star brand EMT gloves (n = 5) withstood the most sandpaper touches before rupturing, with averages of 246 and 245 touches, respectively (Fig 1). Restore Touch and Vglove branded examination gloves were the least durable, withstanding averages of 24.2 and 7.2 touches, respectively (Fig 1). A one-way ANOVA test (α = 0.5) revealed that at least one glove brand had a significantly different average number of touches to failure (P = 0.009). The subsequent Tukey-Kramer HSD test showed that the means from the following brands were significantly different from each other: Curaplex and Vglove (P = 0.0289), Ansell Life Star and Vglove (P = 0.0453), and Curaplex and Restore Touch (P = 0.0487).

The thinnest gloves on average were Restore Touch brand, with a mean of 0.078 mm (Fig 2). The single thinnest glove overall was also a Restore Touch brand glove, with a fingertip thickness of 0.06 mm. This value is barely above the minimum ASTM thickness requirement for nitrile examination gloves, 0.05 mm. All other glove brands measured far exceeded the ASTM minimum (Fig 2). It has previously been reported that Restore Touch and Vglove cuff thickness fell below 0.05 mm (average values of 0.043 mm and 0.048 mm, respectively) [1].

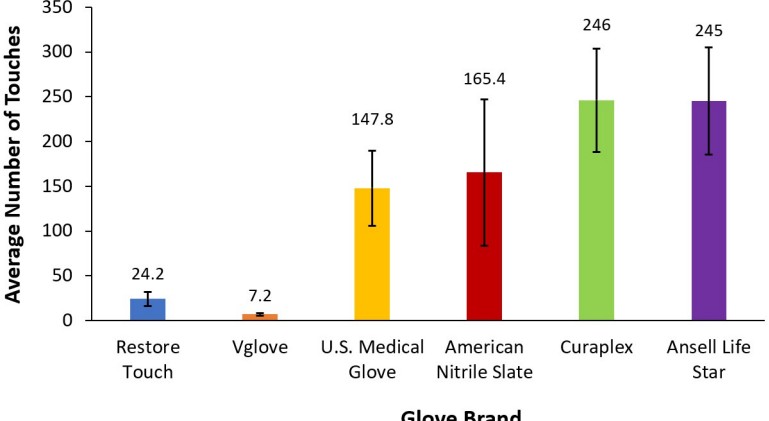

**Fig 1. Number of sandpaper touches to glove failure for Restore Touch, Vglove, U.S. Medical Glove, American Nitrile Slate, Curaplex and Ansell Life Star brands, with n = 5 ± standard error (n = 4 ± standard error for Ansell Life Star due to limited number of samples).**

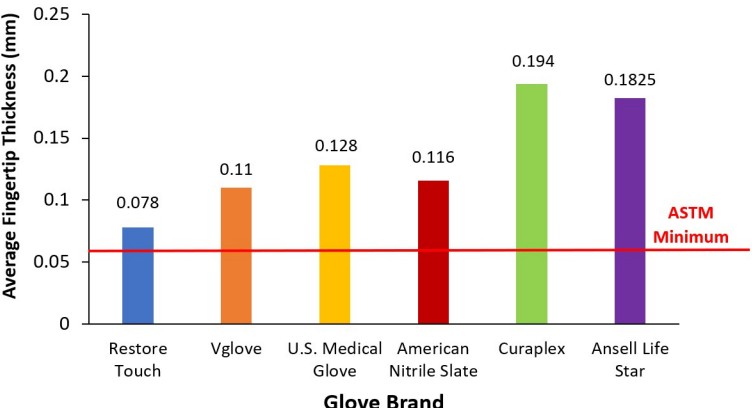

**Fig 2. Average fingertip thickness for Restore Touch, Vglove, U.S. Medical Glove, American Nitrile Slate, Curaplex and Ansell Life Star brands, with n = 5 ± standard error (n = 4 ± standard error for Ansell Life Star).** The horizontal line represents the ASTM minimum acceptable glove thickness for nitrile examination gloves, 0.06 mm.

## Mechanical properties

Evaluation of tensile strength demonstrated that Restore Touch and Vglove had the poorest mechanical properties, with Restore Touch samples breaking between 586% and 697% elongation and Vglove samples between 458% and 632% elongation (Fig 3). In contrast, U.S. Medical Glove samples generally broke above 800% elongation, with the exception of one extreme outlier that broke at only 435% elongation and so was removed from the data set (Fig 3). The American Nitrile Slate brand glove dumbbells broke between 413% and 518% elongation. The two EMT Curaplex and Ansell Life Star glove samples, broke between 450% and 837% and between 574% and 931% elongation, respectively.

## FTIR analysis

Characteristic FTIR peaks of NBR were labeled according to those specified in previous studies of the spectrum of NBR [13–15]. The peak at 2235 cm$^{-1}$ corresponds to the characteristic nitrile (C≡N) group of NBR (Fig 4). Other notable peaks include the = C-H butadiene group at 967 cm$^{-1}$ and -C-H group at 1438 cm$^{-1}$ (Fig 4). These peaks act as a baseline for comparison for the unknown glove samples. Although all of these characteristic peaks are present in the glove samples, the samples also contain peaks that are not present in the FTIR spectrum of pure nitrile, indicating the presence of materials other than NBR (Fig 5).

The FTIR absorbance spectra demonstrate a large amount of overlap, even among glove brands made from different polymers. Natural Hevea latex was included as a reference in order to demonstrate that even a glove made from a natural polymer has very similar FTIR peaks to the synthetic nitrile gloves. The industrial-grade N-Dex Plus glove was included as an additional reference because it is a non-medical nitrile glove. The authors suspect that this large amount of similarity may be due to chemical additives that are universal to the glove manufacturing process. Therefore, the FTIR analyses of the gloves was found to be inconclusive because they did not definitively identify gloves as pure nitrile, as they all claim to be. FTIR appears not to be a suitable tool for definitive glove composition analyses.

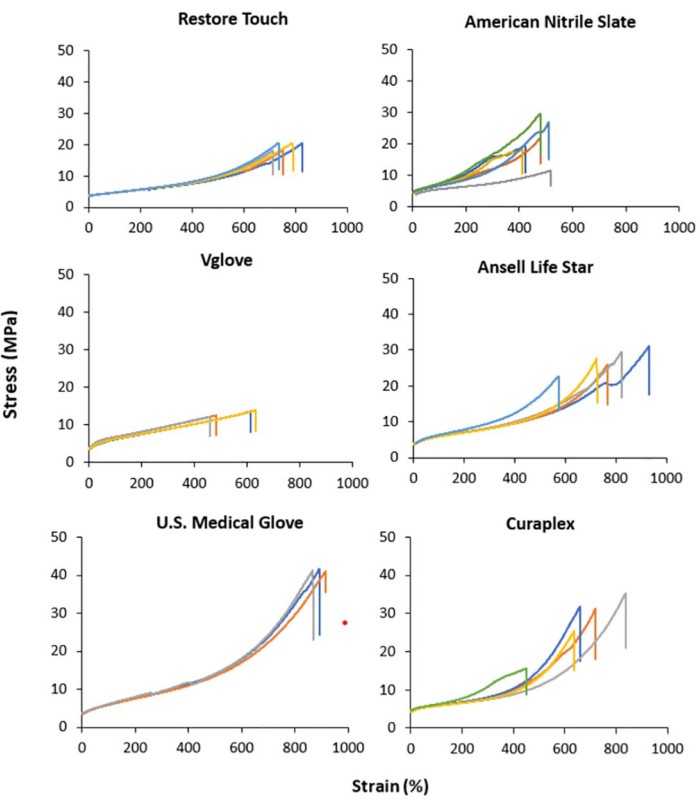

**Fig 3. Tensile stress versus tensile strain for gloves described on their boxes as nitrile examination gloves for 4 glove brands (Restore Touch, American Nitrile Slate, Vglove, and U.S. Medical Glove), and nitrile EMT gloves for 2 glove brands (Ansell Life Star and Curaplex).** Samples were cut using dumbbell Die C. The point at which the plots end is where the testing dumbbell broke in two.

## NMR analysis

The samples consisted of polymers and composite materials that include a variety of functional groups and corresponding chemical shifts (Table 1). Table 1 includes only a selection of the observed chemical shifts. A full list of observed chemical shifts is included in each individual

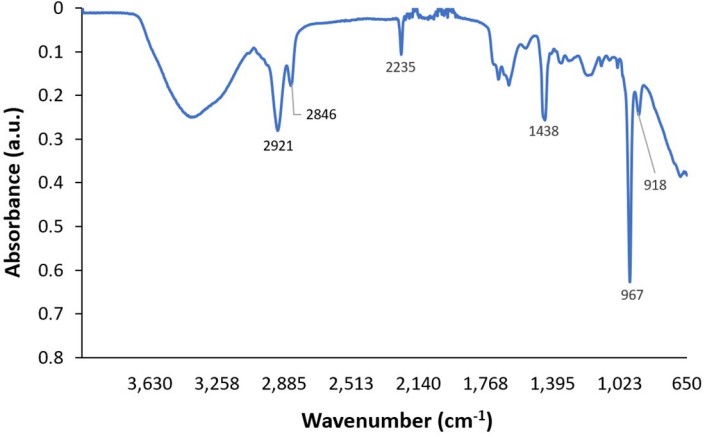

**Fig 4. FTIR absorbance data for pure nitrile with characteristic peaks labeled.**

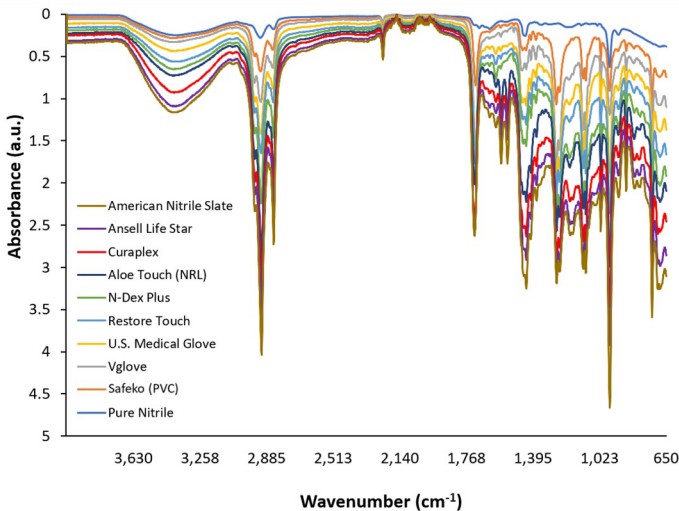

**Fig 5. FTIR absorbance data for Vglove, U.S. Medical Glove, Restore Touch, N-Dex Plus, Curaplex, Ansell Life Star, and American Nitrile Slate brand gloves compared to the positive and negative controls: Pure nitrile and vinyl (PVC), respectively.** A natural latex glove (Hevea latex) was also added as a reference.

spectrum. Multiplicity and J-coupling constants are not observed in these solid-state NMR experiments and are not included in this analysis.

Pure nitrile rubber (dried from an aqueous dispersion of butadiene and acrylonitrile) has clear $^{13}$C chemical shift signature peaks centered around 33.7 ppm and 132.5 ppm, with the latter corresponding to alkenes found in the 1,2 (-C=) polymer and the 1,4 (-C=) polymer (Fig 6). In addition, any C≡N alkyne site would fall within this chemical shift range. This NMR spectral signature of nitrile is found in N-Dex Plus, Restore Touch, U.S. Medical, and all three American Nitrile gloves (Fig 6).

The Safeko PVC glove was analyzed to provide the $^{13}$C chemical shift signature of polyvinyl chloride (PVC) and shows a clearly different $^{13}$C NMR spectrum than that of nitrile, despite overlaps with nitrile rubber due to peaks centered at 131.7 ppm and 33.1 ppm [16]. In PVC, signature peaks of 47.8 ppm and 59.4 ppm can be associated with the CHCl units and the $CH_2$ units of the CHCl-$CH_2$Cl sites of PVC, respectively [17]. Vglove possessed an NMR spectral signature that was nearly identical to Safeko PVC and is therefore consistent with a primarily or entirely PVC composition (Fig 7).

**Table 1. Selection of observed chemical shifts and corresponding functional groups.**

| $^{13}$C Chemical Shift, δ (ppm) | Assignment/Functional Group | Samples |
|---|---|---|
| 33.7 | 1,2 (-C=) nitrile polymer | Pure nitrile, N-Dex Plus, Restore Touch, US Medical, American Nitrile Slate, Curaplex, Ansell Life Star |
| 47.8 | -(-**C**H2–CHCl-)- PVC | Vglove, Safeko PVC |
| 59.4 | -(-CH2–**C**HCl-)- PVC | Vglove, Safeko PVC |
| 131.7 | -C=; –C≡ | Vglove, Safeko PVC |
| 132.5 | 1,4 (-C=) nitrile polymer | Pure nitrile, N-Dex Plus, Restore Touch, US Medical, American Nitrile Slate, Curaplex, Ansell Life Star |
| 132.7 | -C=; –C≡ | Vglove, Safeko PVC |

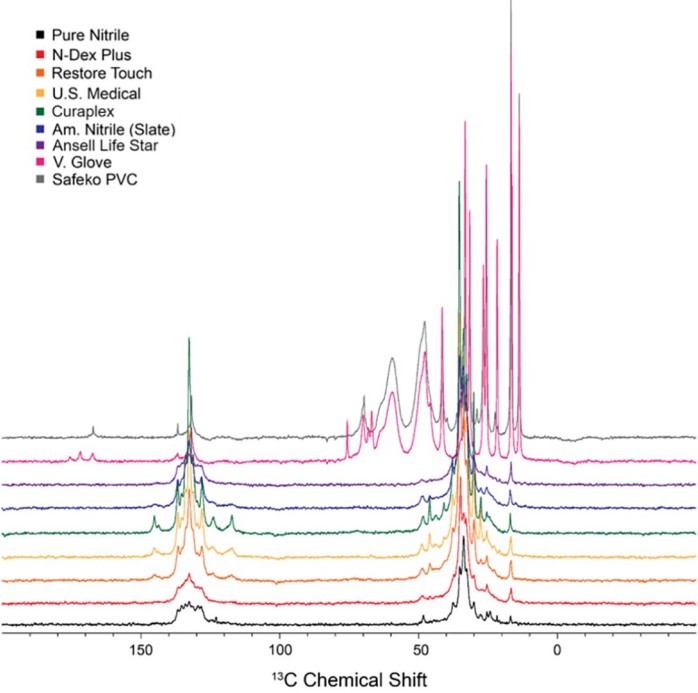

**Fig 6. NMR spectra for Vglove, U.S. Medical Glove, Restore Touch, N-Dex Plus, Curaplex, Ansell Life Star, and American Nitrile Slate brand gloves.** Safeko brand PVC gloves and pure nitrile were also included as positive and negative controls, respectively.

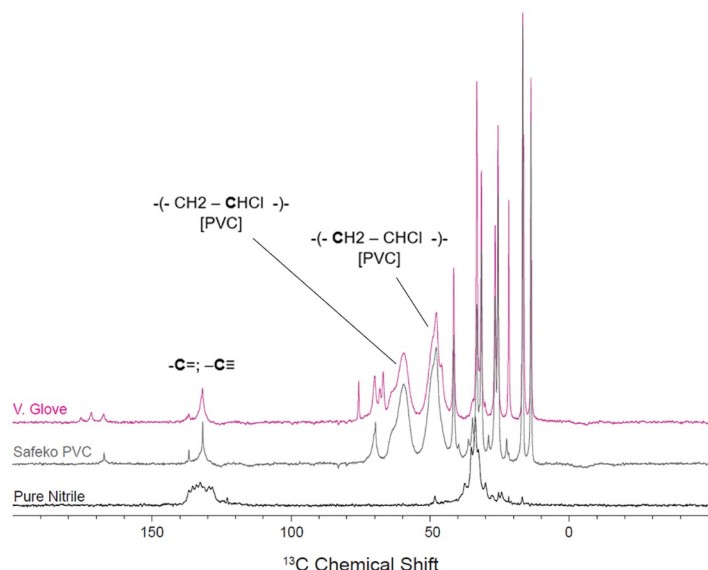

**Fig 7. NMR spectrum for Vglove compared to Safeko PVC and pure nitrile spectra.** Signature peaks of PVC and their corresponding chemical structures are labeled.

**Table 2. Lowest mechanical property values collected for each nitrile glove brand and minimum values specified by ASTM standard D6319.** All tensile strength values collected for the Vglove brand fell below the required minimum of 14 MPa.

| Glove Brand | Tensile Strength (MPa) | Thickness (mm) | Elongation to Break (%) |
|---|---|---|---|
| Restore Touch | 18.03 | 0.06 | 714 |
| Vglove | 12.11 | 0.09 | 458 |
| U.S. Medical Glove | 40.99 | 0.12 | 867 |
| American Nitrile Slate | 18.46 | 0.10 | 413 |
| Curaplex[a] | 35.28 | 0.18 | 837 |
| Ansell Life Star[a] | 22.75 | 0.17 | 574 |
| **ASTM Minimum Value** | **14** | **0.05** | **500** |

[a]Curaplex and Ansell Life Star gloves were included in this table because ASTM International does not have a standard specifically for EMT gloves.

## ASTM standards

ASTM International standard D6319 specifies minimum values for tensile strength, thickness, and elongation to break for nitrile medical examination gloves [9]. Upon comparing the lowest data values collected to the minimum values set by ASTM, Vglove brand glove failed to meet the requirements for both tensile strength (12.11 MPa, compared to a 14 MPa minimum) and elongation to break (458%, compared to 500% minimum) (Table 2). The lowest percent elongation value for American Nitrile Slate gloves also did not reach the 500% elongation minimum, failing at only 413% elongation (Table 2). Two out of the six samples tested exceeded the minimum elongation requirement, while the others did not (Fig 3).

Restore Touch brand gloves, although exceeding the minimum tensile strength and elongation requirements, were barely thicker than the minimum thickness requirements at only 0.06 mm compared to the ASTM minimum of 0.05 mm (Table 2). U.S. Medical Glove brand exceeded all minimum requirements set by ASTM International (Table 2).

A comparison of cost and mechanical properties reveals that only the poor performance Vglove was markedly lower cost than the other gloves (Fig 8). The costs used reflect the purchase cost of the gloves, not the cost of manufacture. We also have not attempted to take into account price savings accrued through bulk purchasing. The Restore Touch glove was slightly cheaper than the U.S. Medical Glove and American Nitrile Slate gloves but was much thinner (Fig 2) suggesting a higher profit margin–at the expense of quality. These two American manufacturers are making much higher performance nitrile gloves than Restore Touch. Their only potential downside is that tactile sensation through these gloves may be less than through the Restore Touch.

## Discussion

The variability in composition and mechanical performance of nitrile medical examination gloves is cause for serious concern. The NMR spectrum of Vglove provides strong evidence that their "nitrile examination glove" is primarily or entirely made from PVC. This mislabeling also explains its inability to meet the minimum requirements set by ASTM International for nitrile examination gloves. Additionally, the lack of manufacturer information on the Vglove packaging should be viewed as a red flag by healthcare purchasers because it makes it much more difficult to hold the manufacturer accountable if their product is substandard. In a previous study in which the durability and mechanical properties of several varieties of medical gloves were compared, PVC gloves were found to be less durable, less stretchy, and less strong than high-quality nitrile gloves [1]. The risks of using PVC gloves in a clinical setting are the

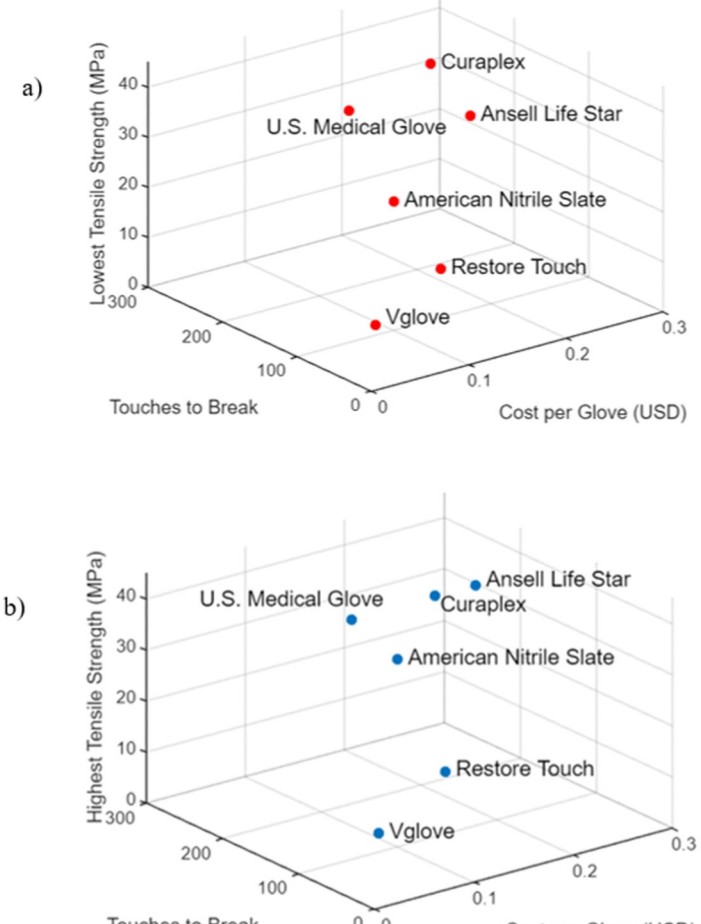

**Fig 8. Comparison of cost per glove, durability (number of touches to break), and a) lowest recorded tensile strength values (MPa) and b) highest recorded tensile strength values (MPa) for Restore Touch, Vglove, U.S. Medical Glove, American Nitrile Slate, Curaplex, and Ansell Life Star.**

same as those for using any variety of flimsy glove: compromised barrier integrity leading to the transfer of diseases and bodily fluids from patient to healthcare provider and vice versa. A company blog post written in 2020 by Elara Brands brought attention to the issue of fraudulent PPE during the COVID-19 pandemic. It stated that the name "Vglove" is a trademark of a legitimate company and that scammers were using the name to sell apparently counterfeit nitrile gloves that were actually made of PVC [18]. These claims are in accordance with the findings of the current study.

Although the NMR spectrum of Restore Touch brand gloves was consistent with that of pure nitrile, the gloves still preformed relatively poorly on the durability test. This poor performance was likely due to the thinness of the gloves, at an average of 0.078 mm and a minimum value of 0.06 mm, which is just above the ASTM minimum requirement of 0.05 mm for nitrile exam gloves [9]. In a previous study, samples of this same glove type and brand were found to have an average cuff thickness less than 0.05, which shows that Restore Touch has also failed to meet ASTM requirements [1]. As shown in Fig 8, it is clear that these very thin gloves fall below the standards, but they are undoubtedly cheaper to make as well as cheaper to buy. Even

a tear the size of a pinhole can allow pathogenic viruses and bacteria, some of which are deadly, to transfer through the medical glove. It is well known that nitrile glove films tear very easily once a break is initiated [1], so even a small rupture can quickly lead to a complete glove failure. This study also demonstrated a positive correlation between glove thickness and durability, which further supports this explanation [1].

The thicker nitrile gloves used by EMTs face the unique challenge of protecting the wearer from exposure to dangerous drugs like fentanyl, a synthetic opioid 50 to 100 times more potent than morphine [19], which is readily absorbed by the human body due to its small size (336.5 Da) [20], in addition to blood-borne pathogens. According to the National Center for Health Statistics at the Center for Disease Control and Prevention, fentanyl and other synthetic opioids are now the most common drugs involved in overdose deaths in the United States, with 70,601 deaths in 2021 [21]. The two brands of EMT gloves evaluated in this study demonstrated both to be suitably strong and durable, although both were quite thick, which reduces tactile sensation by the wearer's hands.

This study has demonstrated that considerable variation in glove quality and performance exists among different brands of nitrile gloves. Although we have not tested all supposed nitrile glove currently in the marketplace, we have proved that Vglove brand nitrile gloves are not made of pure nitrile. This suggests that there may be other nitrile glove brands that are actually made of PVC (Fig 7). The relatively low durability of Restore Touch brand gloves, which are made of nitrile (Fig 6), is because these gloves are very thin, barely meeting minimum thickness requirements set by ASTM International. In contrast, the high-performance nitrile examination gloves made by US Medical Glove were almost as strong and durable as the thick EMT gloves. It is likely that other properties of the gloves such as heat resistance, oxygen aging resistance, solvent resistance also vary. However, these only become relevant in a glove that remains intact during normal use, so were not tested here.

Future studies will include analyses of the mechanical performance and NMR spectra of a wider variety of nitrile medical glove brands to gain a more complete understanding of the quality of gloves on the market. Data on additional glove properties, such as stress relaxation and solvent penetration rates, may also be collected in a future study. An analysis of the durability of gloves exposed to common solvents will also be conducted.

## Conclusions

The marketing of substandard gloves can have serious consequences for healthcare workers who rely on personal protective equipment such as medical gloves to protect themselves and their patients from healthcare-associated infections (HAIs). Therefore, gloves that fail to meet even the minimum requirements set by ASTM International and the FDA should not be used for medical purposes. A new standard requiring glove manufacturers to disclose the durability of their gloves on their packaging has been proposed to ASTM International and is currently under review. These label changes would enable healthcare workers to more easily identify and avoid substandard gloves. In addition, increasing inspection rates and raising the penalty for violating manufacturing standards would deter glove manufacturers from selling low-quality medical gloves. The authors urge those who regularly use nitrile examination gloves to emphasize glove quality over other considerations to adequately protect themselves and their patients.

## Supporting information

**S1 File. Solid state NMR spectra and data for each sample.**
(PDF)

**S2 File. Raw NMR data files.**
(ZIP)

## Acknowledgments

The authors thank EnergyEne, Inc. for permission to use the GAD device in this study. We also thank U.S. Medical Glove and American Nitrile for generously donating their gloves to be used this study.

## Author Contributions

**Conceptualization:** Ashley Herkins, Katrina Cornish.

**Data curation:** Ashley Herkins, Sriloy Dey, Dan Conroy.

**Formal analysis:** Ashley Herkins, Dan Conroy.

**Funding acquisition:** Katrina Cornish.

**Investigation:** Ashley Herkins, Sriloy Dey, Dan Conroy.

**Methodology:** Ashley Herkins, Sriloy Dey, Dan Conroy.

**Project administration:** Katrina Cornish.

**Software:** Sriloy Dey, Dan Conroy.

**Supervision:** Katrina Cornish.

**Validation:** Ashley Herkins.

**Visualization:** Ashley Herkins.

**Writing – original draft:** Ashley Herkins.

**Writing – review & editing:** Katrina Cornish.

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
