## [Decision Letter · Decision Letter 0]

19 Jul 2024

PONE-D-24-03435Nitrile Glove Composition and Performance – Substandard Properties and Inaccurate Packaging InformationPLOS ONE

Dear Dr. Cornish,

Thank you for submitting your manuscript to PLOS ONE. After careful consideration, we feel that it has merit but does not fully meet PLOS ONE’s publication criteria as it currently stands. Therefore, we invite you to submit a revised version of the manuscript that addresses the points raised during the review process.

Please see the comments from two reviewers below. We apologise for the delay in getting this decision to you, as the original Academic Editor became unavailable. Please be aware that the editor who handles your revised manuscript might find it necessary to invite additional reviewers to assess this work once the revised manuscript is submitted. Please also provide additional information on how the statistical analysis was carried out, in sufficient detail for another researcher to be able to replicate the analysis.

We look forward to receiving your revised manuscript.

Kind regards,

Hanna Landenmark

Staff Editor

PLOS ONE

2. We note that this submission includes NMR spectroscopy data. We would recommend that you include the following information in your methods section or as Supporting Information files:

a) The make/source of the NMR instrument used in your study, as well as the magnetic field strength. For each individual experiment, please also list: the nucleus being measured; the sample concentration; the solvent in which the sample is dissolved and if solvent signal suppression was used; the reference standard and the temperature.

b) A list of the chemical shifts for all compounds characterised by NMR spectroscopy, specifying, where relevant: the chemical shift (δ), the multiplicity and the coupling constants (in Hz), for the appropriate nuclei used for assignment.

c)The full integrated NMR spectrum, clearly labelled with the compound name and chemical structure.

We also strongly encourage authors to provide primary NMR data files, in particular for new compounds which have not been characterised in the existing literature. Authors should provide the acquisition data, FID files and processing parameters for each experiment, clearly labelled with the compound name and identifier, as well as a structure file for each provided dataset. See our list of recommended repositories here: https://journals.plos.org/plosone/s/recommended-repositories

“This work was supported by the USDA National Institute of Food and Agriculture, OHO01524, accession 7003189.”

“I have read the journal's policy and the authors of this manuscript have the following competing interests: Katrina Cornish is the CEO of EnergyEne, Inc. the company that lent the GAD for use in this study. The other authors declare no conflicts of interest.”

5. In this instance it seems there may be acceptable restrictions in place that prevent the public sharing of your minimal data. However, in line with our goal of ensuring long-term data availability to all interested researchers, PLOS’ Data Policy states that authors cannot be the sole named individuals responsible for ensuring data access (http://journals.plos.org/plosone/s/data-availability#loc-acceptable-data-sharing-methods).

7. Please ensure that you include a title page within your main document. You should list all authors and all affiliations as per our author instructions and clearly indicate the corresponding author.

Reviewers' comments:

Reviewer's Responses to Questions

**Comments to the Author**

1. Is the manuscript technically sound, and do the data support the conclusions?

Reviewer #1: Yes

Reviewer #2: Yes

2. Has the statistical analysis been performed appropriately and rigorously? 

Reviewer #1: N/A

Reviewer #2: Yes

3. Have the authors made all data underlying the findings in their manuscript fully available?

Reviewer #1: No

Reviewer #2: Yes

4. Is the manuscript presented in an intelligible fashion and written in standard English?

Reviewer #1: Yes

Reviewer #2: Yes

5. Review Comments to the Author

Reviewer #1: The manuscript entitled "Nitrile Glove Composition and Performance – Substandard Properties and Inaccurate Packaging Information" by Cornish et al is very interesting and significant But the following concerns needed to be addressed before it can be published in PLOS ONE.

1. Is there any allergic reactions performed for this gloves? If not needed please give explanations

2. What about the flexibility of these gloves? As I have observed Nitrile Gloves are sometimes less flexible

3. What are the cost analysis off these gloves?

4. These gloves are biodegradable?

5. Is there any exceptional chemical present which could be penetrate these gloves?

6. Is these gloves are temperature sensitive?

Reviewer #2: This study evaluates the composition, durability, and mechanical properties of various nitrile glove brands. The authors found significant variability in glove performance, with some brands failing to meet ASTM standards despite being marketed as nitrile. NMR analysis revealed that one brand, Vglove, had a spectral signature consistent with PVC rather than nitrile. Thinner gloves like Restore Touch showed poor durability despite being composed of nitrile. The authors make a compelling case that glove quality should be prioritized to ensure healthcare worker and patient safety. The manuscript is well-organized and the methods are sound, but some areas need further clarification or development. Expanding the sample size and variety of glove brands tested would strengthen the conclusions. With some revisions to address the comments below, this paper has the potential to make a valuable contribution to our understanding of medical glove performance and composition.

Comments:

1. The introduction provides good context on the importance of glove integrity for healthcare worker safety. However, it would be helpful to include more background information on the specific chemical and physical properties of nitrile that make it a preferred material for medical gloves.

2. In the methods section, please clarify how the glove thickness measurements were taken. Were multiple spots measured on each glove to account for potential variations in thickness?

3. The FTIR results showed a high degree of spectral overlap between the glove samples, even those made of different materials. What are some potential reasons for this similarity, and how might it impact the utility of FTIR for glove composition analysis?

4. The NMR data provide compelling evidence that the Vglove brand is primarily composed of PVC rather than nitrile. Are there any other analytical techniques that could be used to further validate this finding?

5. How do the tensile strength and elongation values for the EMT gloves compare to the ASTM D6319 requirements for nitrile examination gloves? It would be informative to include this context even though there is not a specific EMT glove standard.

6. The authors note that future studies will analyze a wider variety of nitrile glove brands. Will these studies also include gloves marketed as latex or vinyl to assess their composition and performance relative to nitrile?

7. In the conclusion, the authors emphasize that gloves failing to meet minimum ASTM requirements should not be used for medical purposes. How can healthcare facilities and practitioners ensure they are purchasing high-quality gloves that meet these standards?

8. The durability testing used 120-grit sandpaper to create a rough contact surface. Is this grit size representative of the abrasive surfaces gloves might encounter in a clinical setting?

9. Table 1 compares the lowest mechanical property values to ASTM standards. Would it also be valuable to report the average values to give a more complete picture of each glove's performance?

10. The cost and mechanical properties comparison in Figure 8 is informative. Have the authors considered normalizing the cost per glove to account for variations in glove thickness?

11. In the introduction, the authors mention that low inspection rates during the pandemic allowed poorly made gloves to enter the US market. Do they have any recommendations for improving glove quality control and regulation going forward?

12. The Vglove brand did not provide any manufacturing information on their packaging or website. Is this lack of transparency a potential red flag that healthcare purchasers should be aware of?

13. Could variations in glove storage conditions or age impact their mechanical properties or durability? If so, how was this accounted for in the study design?

14. The discussion notes that PVC gloves have been found to be less durable and strong than high-quality nitrile gloves. What are some of the specific risks associated with using PVC exam gloves in clinical settings?

15. Beyond tensile strength and durability, what other glove properties might be important to assess in future studies to get a more complete picture of their overall quality and performance?

6. PLOS authors have the option to publish the peer review history of their article (what does this mean?). If published, this will include your full peer review and any attached files.

Reviewer #1: **Yes: **Ihtisham Ul Haq

Reviewer #2: No

---

## [Author Response · Author response to Decision Letter 0]

6 Sep 2024

The manuscript has been re-formatted to meet PLOS ONE’s style requirements.

2. We note that this submission includes NMR spectroscopy data. We would recommend that you include the following information in your methods section or as Supporting Information files:

a) The make/source of the NMR instrument used in your study, as well as the magnetic field strength. For each individual experiment, please also list: the nucleus being measured; the sample concentration; the solvent in which the sample is dissolved and if solvent signal suppression was used; the reference standard and the temperature.

The following information was included in the “Solid State Nuclear Magnetic Resonance Tests (NMR)” subsection of the Methods section: 

Bruker Avance IIIHD 600MHz (14.1T) DNP-NMR spectrometer equipped with a 3.2mm triple-resonance (HXY) DNP probe tuned in 1H-13C double mode. 

All samples were packed into 3.2mm zirconium rotors and spun at 10kHz MAS at 300K. All spectra are 13C quantitative CP experiments with 11ms total CP duration and 10k scans were calibrated on an external standard (N-acetyl-valine).

Lines 154-155 were added for clarity: “Note that these were solid samples. They were not dissolved in any solvent and therefore do not have a concentration to reference. No solvent suppression was used.”

b) A list of the chemical shifts for all compounds characterized by NMR spectroscopy, specifying, where relevant: the chemical shift (δ), the multiplicity and the coupling constants (in Hz), for the appropriate nuclei used for assignment.

Lines 227-231 were added for additional clarity: “The samples consisted of polymers and composite materials that include a variety of functional groups and corresponding chemical shifts (Table 1). Table 1 includes only a selection of the observed chemical shifts. A full list of observed chemical shifts is included in each individual spectrum. Multiplicity and J-coupling constants are not observed in these solid-state NMR experiments and are not included in this analysis.”

Table 1 was also created to summarize the chemical shifts and corresponding functional groups (Lines 233-234).

13C Chemical Shift, δ (ppm) Assignment/Functional Group Samples

33.7 1,2 (-C=) nitrile polymer Pure nitrile, N-Dex Plus, Restore Touch, US Medical, American Nitrile Slate, Curaplex, Ansell Life Star

47.8 -(-CH2–CHCl-)- PVC Vglove, Safeko PVC

59.4 -(-CH2–CHCl-)- PVC Vglove, Safeko PVC

131.7 -C=; –C≡ Vglove, Safeko PVC

132.5 1,4 (-C=) nitrile polymer Pure nitrile, N-Dex Plus, Restore Touch, US Medical, American Nitrile Slate, Curaplex, Ansell Life Star

132.7 -C=; –C≡ Vglove, Safeko PVC

c) The full integrated NMR spectrum, clearly labelled with the compound name and chemical structure. We also strongly encourage authors to provide primary NMR data files, in particular for new compounds which have not been characterised in the existing literature. Authors should provide the acquisition data, FID files and processing parameters for each experiment, clearly labelled with the compound name and identifier, as well as a structure file for each

provided dataset. See our list of recommended repositories here: https://journals.plos.org/plosone/s/recommendedrepositories

See the attached spectra:

• American Nitrile (green)

• American Nitrile (purple)

• American Nitrile (slate)

• N-Dex Plus

• Pure nitrile

• Restore Touch

• Safeco PVC

• U.S. Medical

• Vglove

The acquisition settings and parameters for all spectra are identical: 13C quantitative CP experiments acquired at 10kHz MAS and 300K with 11ms total CP duration. Recycle delays are set to 3.0 s and each spectrum is acquired with 10240 scans. Spectra widths are 394 ppm with an acquisition time of 34.4ms. Spectra were processed in Bruker TopSpin with no additional linebroadening and with a polynomial baseline correction.

To clarify these additional details, lines 151-154 were added to the manuscript: “Recycle delays were set to 3.0 s. Spectra widths were 394 ppm with an acquisition time of 34.4ms. Spectra were processed in Bruker TopSpin with no additional line broadening and with a polynomial baseline correction. The acquisition settings and parameters for all spectra were identical.”

Note: I was not able to open the link to ‘recommended repositories’ to double check the content/format.

“This work was supported by the USDA National Institute of Food and Agriculture, OHO01524, accession 7003189.” Please state what role the funders took in the study. If the funders had no role, please state: "The funders had no role in study design, data collection and analysis, decision to publish, or preparation of the manuscript." If this statement is not correct you must amend it as needed. Please include this amended Role of Funder statement in your cover letter; we will change the online submission form on your behalf.

Lines 366-367 were added to the manuscript: “The funders had no role in study design, data collection and analysis, decision to publish, or preparation of the manuscript.”

“I have read the journal's policy and the authors of this manuscript have the following competing interests: Katrina Cornish is the CEO of EnergyEne, Inc. the company that lent the GAD for use in this study. The other authors declare no conflicts of

interest.”

Please confirm that this does not alter your adherence to all PLOS ONE policies on sharing data and materials, by including the following statement: "This does not alter our adherence to PLOS ONE policies on sharing data and materials.” (as detailed online in our guide for authors http://journals.plos.org/plosone/s/competing-interests). If there are restrictions on sharing of data and/or materials, please state these. Please note that we cannot proceed with consideration of your article until this information has been declared. Please include your updated Competing Interests statement in your cover letter; we will change the online submission form on your behalf.

The updated competing interest statement has been included in the cover letter: “Also, we (the authors) have read the journal's policy and have the following competing interests: Katrina Cornish is the CEO of EnergyEne, Inc. the company that lent the GAD for use in this study. This does not alter our adherence to PLOS ONE policies on sharing data and materials. The other authors declare no conflicts of interest.”

5. In this instance it seems there may be acceptable restrictions in place that prevent the public sharing of your minimal data. However, in line with our goal of ensuring long-term data availability to all interested researchers, PLOS’ Data Policy states that authors cannot be the sole named individuals responsible for ensuring data access (http://journals.plos.org/plosone/s/data-availability#loc-acceptable-data-sharing-methods).

Data requests to a non-author institutional point of contact, such as a data access or ethics committee, helps guarantee long term stability and availability of data. Providing interested researchers with a durable point of contact ensures data will be accessible even if an author changes email addresses, institutions, or becomes unavailable to answer requests. Before we proceed with your manuscript, please also provide non-author contact information (phone/email/hyperlink) for a data access committee, ethics committee, or other institutional body to which data requests may be sent. If no institutional body is available to respond to requests for your minimal data, please consider if there any institutional representatives

who did not collaborate in the study, and are not listed as authors on the manuscript, who would be able to hold the data and respond to external requests for data access? If so, please provide their contact information (i.e., email address). Please also provide details on how you will ensure persistent or long-term data storage and availability.

Will Ray is the designated contact for data storage for the Department of Food, Agricultural, and Biological Engineering at Ohio State. He is responsible for ensuring persistent and long-term storage of data. His contact is ray.29@osu.edu.

6. PLOS requires an ORCID iD for the corresponding author in Editorial Manager on papers submitted after December 6th, 2016. Please ensure that you have an ORCID iD and that it is validated in Editorial Manager. To do this, go to ‘Update my Information’ (in the upper left-hand corner of the main menu), and click on the Fetch/Validate link next to the ORCID field.

This will take you to the ORCID site and allow you to create a new iD or authenticate a pre-existing iD in Editorial Manager. Please see the following video for instructions on linking an ORCID iD to your Editorial Manager account: https://www.youtube.com/watch?v=_xcclfuvtxQ

The corresponding author has linked her ORCID iD. 

7. Please ensure that you include a title page within your main document. You should list all authors and all affiliations as per our author instructions and clearly indicate the corresponding author.

A title page listing all authors and affiliations has been added to the main document. 

Reviewer #1: The manuscript entitled "Nitrile Glove Composition and Performance – Substandard Properties and Inaccurate Packaging Information" by Cornish et al is very interesting and significant, but the following concerns needed to be addressed before it can be published in PLOS ONE.

1. Is there any allergic reactions performed for this gloves? If not needed please give explanations.

Lines 53-55 state the following regarding allergic reactions: “Although not as durable or as comfortable as natural latex gloves [1-5] nitrile and other synthetic gloves do not induce Type I latex allergic reactions, as may occur when using improperly leached natural latex protects [6].”

Lines 55-57 were added for additional clarity: “Residual chemicals may be present at different levels in the various glove lots which could cause contact reactions. However, it is not the purview of this paper to ensure compliance with all the FDA requirements for nitrile gloves.”

2. What about the flexibility of these gloves? As I have observed Nitrile Gloves are sometimes less flexible

Lines 127-129 were added to the manuscript: “The stress and strain data collected also provides information on the modulus of elasticity, or stiffness of the glove. The modulus of each glove type is not explicitly stated, however, for brevity.”

3. What are the cost analysis off these gloves?

Lines 276-284 include a cost analysis of the gloves. 

4. These gloves are biodegradable?

Line 58 was added to address this question: “Because nitrile is derived from petroleum, it is also a non-biodegradable material.”

5. Is there any exceptional chemical present which could be penetrate these gloves?

None of our tests included chemicals which might be expected to penetrate the gloves. We only considered “normal conditions” in this study. However, we have published a study (https://doi.org/10.1186/s13037-024-00400-4) where we explored the effects of different common liquids on glove durability. Chemical penetration of nitrile gloves is beyond the scope of the study. The general barrier integrity of the gloves under different conditions was analyzed using the GAD. There is future scop for testing chemotherapy drugs or preservatives used by morticians if glove companies wish to explore this aspect further, but such chemical testing is beyond our current scope of work. 

6. Is these gloves are temperature sensitive?

Lines 85-86 were added for clarity: “Because glove properties can vary depending on temperature, all gloves were stored and tested at room temperature (22 °C).” We hope investigators may be interested in exploring this aspect of glove safety and durability in the future.

Reviewer #2: This study evaluates the composition, durability, and mechanical properties of various nitrile glove brands. The authors found significant variability in glove performance, with some brands failing to meet ASTM standards despite being marketed as nitrile. NMR analysis revealed that one brand, Vglove, had a spectral signature consistent with PVC rather than nitrile. Thinner gloves like Restore Touch showed poor durability despite being composed of nitrile. The authors make a compelling case that glove quality should be prioritized to ensure healthcare worker and patient safety. The manuscript is well-organized and the methods are sound, but some areas need further clarification or development. Expanding the sample size and variety of glove brands tested would strengthen the conclusions. With some revisions to

address the comments below, this paper has the potential to make a valuable contribution to our understanding of medical glove performance and composition.

Comments:

1. The introduction provides good context on the importance of glove integrity for healthcare worker safety. However, it would be helpful to include more background information on the specific chemical and physical properties of nitrile that make it a preferred material for medical gloves.

The following information is found in the Introduction section of the manuscript: 

Among synthetic elastomers, nitrile has gained wide acceptance as a glove that has much better performance (strength, softness and elasticity) than PVC or PE gloves, but is much cheaper than the better performing polyisoprene and chloroprene gloves. However, although nitrile gloves have improved in mechanical properties since their introduction, they are not generally viewed as a preferred glove material overall because they lack the properties of natural latex gloves, and high cost synthetics, and have minimal tear resistance. However, the gloves are protein allergen-free.

2. In the methods section, please clarify how the glove thickness measurements were taken. Were multiple spots measured on each glove to account for potential variations in thickness?

Lines 116-119 were modified to include a more detailed description of the procedure for glove thickness measurements. The lines now read, “Following durability testing, the middle finger of each glove was removed, the thickness of the fingertip (mm) was measured three times using electronic calipers, and the median value was recorded. The average of these values was then reported for each glove variety.”

3. The FTIR results showed a high degree of spectral overlap between the glove samples, even those made of different materials. What are some potential reasons for this similarity, and how might it impact the utility of FTIR for glove composition analysis?

Lines 220-222 explain the likeliest reason for the overlap: “The authors suspect that this large amount of similarity may be due to chemical additives that are universal to the glove manufacturing process.”

Lines 223-224 were added for additional clarity: “FTIR appears not to be a suitable tool for definitive glove composition analyses.”

4. The NMR data provide compelling evidence that the Vglove brand is primarily composed of PVC rather than nitrile. Are there any other analytical techniques that could be used to further validate this finding?

NMR was found to be the most well-suited technique for glove composition analysis because of its precision. However, the findings from the NMR analysis were also validated by the fact that the Vglove failed to meet the minimum values for tensile strength, thickness, and elongation to break for nitrile medical examination gloves set by ASTM International. 

5. How do the tensile strength and elongation values for the EMT gloves compare to the ASTM D6319 requirements for

---

## [Decision Letter · Decision Letter 1]

16 Oct 2024

Nitrile Glove Composition and Performance – Substandard Properties and Inaccurate Packaging Information

PONE-D-24-03435R1

Dear Dr. Cornish,

We’re pleased to inform you that your manuscript has been judged scientifically suitable for publication and will be formally accepted for publication once it meets all outstanding technical requirements.

Kind regards,

Mohamed Moafak Arbili

Academic Editor

PLOS ONE

Additional Editor Comments (optional):

The manuscript titled "Nitrile Glove Composition and Performance – Substandard Properties and Inaccurate Packaging Information" has been well revised. It is now ready to proceed to the next stage of publication.

Reviewers' comments:

Reviewer's Responses to Questions

**Comments to the Author**

1. If the authors have adequately addressed your comments raised in a previous round of review and you feel that this manuscript is now acceptable for publication, you may indicate that here to bypass the “Comments to the Author” section, enter your conflict of interest statement in the “Confidential to Editor” section, and submit your "Accept" recommendation.

Reviewer #1: All comments have been addressed

Reviewer #2: All comments have been addressed

2. Is the manuscript technically sound, and do the data support the conclusions?

Reviewer #1: Yes

Reviewer #2: Yes

3. Has the statistical analysis been performed appropriately and rigorously? 

Reviewer #1: N/A

Reviewer #2: Yes

4. Have the authors made all data underlying the findings in their manuscript fully available?

Reviewer #1: Yes

Reviewer #2: Yes

5. Is the manuscript presented in an intelligible fashion and written in standard English?

Reviewer #1: Yes

Reviewer #2: Yes

6. Review Comments to the Author

Reviewer #1: The manuscript is significantly improved, Authors addresses all the comments seriously, thus i recommend accept

Reviewer #2: Well revised. "Nitrile Glove Composition and Performance – Substandard Properties and Inaccurate Packaging Information" Can be processed for the next stage of publication.

7. PLOS authors have the option to publish the peer review history of their article (what does this mean?). If published, this will include your full peer review and any attached files.

Reviewer #1: **Yes: **Ihtisham Ul Haq

Reviewer #2: **Yes: **Mahmoud H. Akeed

---

## [Editor Report · Acceptance letter]

20 Oct 2024

PONE-D-24-03435R1 

PLOS ONE

Dear Dr. Cornish, 

I'm pleased to inform you that your manuscript has been deemed suitable for publication in PLOS ONE. Congratulations! Your manuscript is now being handed over to our production team.

Kind regards, 

on behalf of

Dr. Mohamed Moafak Arbili 

Academic Editor

PLOS ONE